# Timeseries Anomaly Detection using Temporal Hierarchical One-Class Network

**Lifeng Shen[1], Zhuocong Li[2], James T. Kwok[1]**
[1] Department of Computer Science and Engineering,
Hong Kong University of Science and Technology, Hong Kong
{lshenae,jamesk}@cse.ust.hk
[2] Cloud and Smart Industries Group, Tencent, China
zhuocongli@tencent.com

## Abstract

Real-world timeseries have complex underlying temporal dynamics and the detection of anomalies is challenging. In this paper, we propose the Temporal Hierarchical One-Class (THOC) network, a temporal one-class classification model for timeseries anomaly detection. It captures temporal dynamics in multiple scales by using a dilated recurrent neural network with skip connections. Using multiple hyperspheres obtained with a hierarchical clustering process, a one-class objective called Multiscale Vector Data Description is defined. This allows the temporal dynamics to be well captured by a set of multi-resolution temporal clusters. To further facilitate representation learning, the hypersphere centers are encouraged to be orthogonal to each other, and a self-supervision task in the temporal domain is added. The whole model can be trained end-to-end. Extensive empirical studies on various real-world timeseries demonstrate that the proposed THOC network outperforms recent strong deep learning baselines on timeseries anomaly detection.

## 1 Introduction

In complex cyber-physical systems such as power plants, data centers and smart factories, there are tons of sensors operating and generating substantial amounts of measurements continuously. To help monitor the system's real-time working conditions, it is critical to be able to find anomalies such that potential risks and financial losses can be avoided. The problem of identifying the system's abnormal status in each time step of the timeseries data is called timeseries anomaly detection [31]. A comprehensive survey on the traditional techniques can be found in [6].

An effective timeseries anomaly detection method should be able to model the complex nonlinear temporal dynamics of the underlying system's normal behavior, while being robust and can be generalized to unseen anomalies. However, the development of such a technique is very challenging. First, real-world timeseries have highly nonlinear temporal dependencies and complex interactions among variables. Moreover, anomalies are often rare. The finding and labeling of these anomalies are very time-consuming and expensive in practice. Hence, timeseries anomaly detection is usually formulated in the unsupervised learning setting [28], which is also the focus in this paper.

One-class classification [19] is a popular paradigm for anomaly detection. The idea is that by assuming that most of the training data are normal, their characteristics are captured and learned by a model. An outlier is then detected when the current observation cannot be well-fitted by the model. Two well-known and closely related one-class classification models are the one-class support vector machine (OC-SVM) [26], which uses a hyperplane to separate the normal data from anomalous data; and the support vector data description (SVDD) [29], which uses a hypersphere to enclose the normal

data. While they have been successfully used in many real-world applications [3, 15, 32], they are often limited to data-rich scenarios so that the normal patterns can be sufficiently captured.

The OC-SVM and SVDD rely on the kernel trick [27] to map input features to a high-dimensional space for data separation. Motivated by the immense success of deep learning in various applications such as computer vision, speech recognition and natural language processing [12, 25], recent efforts try to integrate the powerful representation learning ability of deep networks into the traditional one-class classifiers. For example, deep SVDD [22] replaces the kernel-induced feature space in SVDD by the feature space learned in a deep network. DAGMM [34] is a density-based one-class classifier that integrates a deep autoencoder with the Gaussian mixture model (GMM), such that the normal data can be well-captured by the GMM in a low-dimensional latent space. As in other deep networks, these models can all be conveniently trained via back-propagation in an end-to-end manner.

The above-mentioned traditional/deep one-class classifiers are designed for fixed-dimensional input data. It is still an open issue on how to extend them for timeseries anomaly detection. A simple approach is to run a sliding window on the timeseries data. A fixed-dimensional vector containing the history information is then extracted and fed to the one-class classifier. However, this fails to adequately capture the underlying temporal dependency. To alleviate this problem, a number of models based on recurrent networks have been recently proposed for timeseries anomaly detection. An early attempt is a LSTM-based encoder-decoder model [17], and an anomaly score is defined based on the reconstruction error on the timeseries. However, it suffers from error accumulation on decoding a long sequence. Other more powerful deep generative models, such as the recurrent variational autoencoder [28], and variants of the generative adversarial network (GAN) (e.g., MAD-GAN [13] and BeatGAN [33]) have also been proposed. However, training of the GAN is usually difficult, and requires a careful balance between the discriminator and generator [10].

Inspired by deep SVDD, we propose in this paper the Temporal Hierarchical One-Class (THOC) network. First, it uses the dilated recurrent neural network (RNN) [2] with skip connections to efficiently extract multi-scale temporal features from the timeseries. Instead of using only the lowest-resolution features obtained at the top layer of the dilated RNN, THOC fuses features from all intermediate layers together by a differentiable hierarchical clustering mechanism. At each resolution, normal behaviors are represented by multiple hyperspheres. This captures the complex characteristics in real-world timeseries data, and is more powerful than the use of a single hypersphere in deep SVDD. A multiscale support vector data description (MVDD), which is a one-class objective defined based on the difference between the fused multiscale features and hypersphere centers, allows the whole model to be trained end-to-end. Finally, a novelty score, which measures how the current observation deviates from the normal behaviors represented by the hyperspheres, is used for anomaly detection on an unseen observation. Experiments performed on a number of real-world timeseries data sets show that the proposed model outperforms the recent state-of-the-arts.

## 2   Related Work

In this section, we briefly review the support vector data description (SVDD) [29], and the more recent deep SVDD [22]. Given a set of $N$ data samples $\{\mathbf{x}_1, \ldots, \mathbf{x}_N\}$, in which most of them are normal but some are anomalous (outliers), SVDD tries to find a small hypersphere (with center $\mathbf{c}$ and radius $R$) to enclose the normal data. This can be formulated as the following optimization problem:

$$\min_{\mathbf{c},R,\xi} \quad R^2 + \frac{1}{\nu N} \sum_{i=1}^{N} \xi_i \tag{1}$$

$$\text{s.t.} \quad \|\phi(\mathbf{x}_i) - \mathbf{c}\|^2 \leq R^2 + \xi_i, \ \xi_i \geq 0 \quad \forall i = 1, \ldots, N, \tag{2}$$

where $\phi$ is a kernel-induced feature map.

Deep SVDD improves SVDD by replacing $\phi(\cdot)$ with representations learned by a deep network. Analogous to (1), its optimization problem becomes

$$\min_{R,\mathcal{W}} \ R^2 + \frac{1}{\nu N} \sum_{i=1}^{N} \max\{0, \|\text{NN}(\mathbf{x}_i; \mathcal{W}) - \mathbf{c}\|^2 - R^2\} + \lambda \Omega(\mathcal{W}),$$

where $\mathrm{NN}(\cdot;\mathcal{W})$ is a deep network with parameter $\mathcal{W}$, and $\Omega(\mathcal{W})$ is a regularizer (such as the $\ell_2$-regularizer). Empirically, it is found that the following simplified objective yields better performance:

$$\min_{\mathcal{W}} \frac{1}{N} \sum_{i=1}^{N} \|\mathrm{NN}(\mathbf{x}_i; \mathcal{W}) - \mathbf{c}\|^2 + \lambda \Omega(\mathcal{W}). \tag{3}$$

The whole model is then learned end-to-end.

## 3   Temporal Hierarchical One-Class (THOC) Network

In timeseries anomaly detection, we are given a set of timeseries $\mathcal{D} = \{\mathbf{X}_1, \ldots, \mathbf{X}_N\}$. $\mathbf{X}_s$ is of length $T_s$, and its observation at time $t$ is $\mathbf{x}_{t,s} \in \mathbb{R}^D$. The task is to determine if $\mathbf{x}_{t,s}$ is anomalous, based on the partial timeseries $\mathbf{x}_{1:t,s}$ that have been observed so far. Following [17, 28, 13, 31], we consider the unsupervised learning setting, and do not use any label information during training. This is more practical as labeled anomalies are rare and often difficult to identify.

### 3.1   Architecture

Figure 1 shows the proposed architecture. On the left, temporal features at multiple time scales are extracted from the timeseries (Section 3.1.1). On the right, the features are fused and processed by a hierarchical network, which outputs an anomaly score at the top (Section 3.1.2).

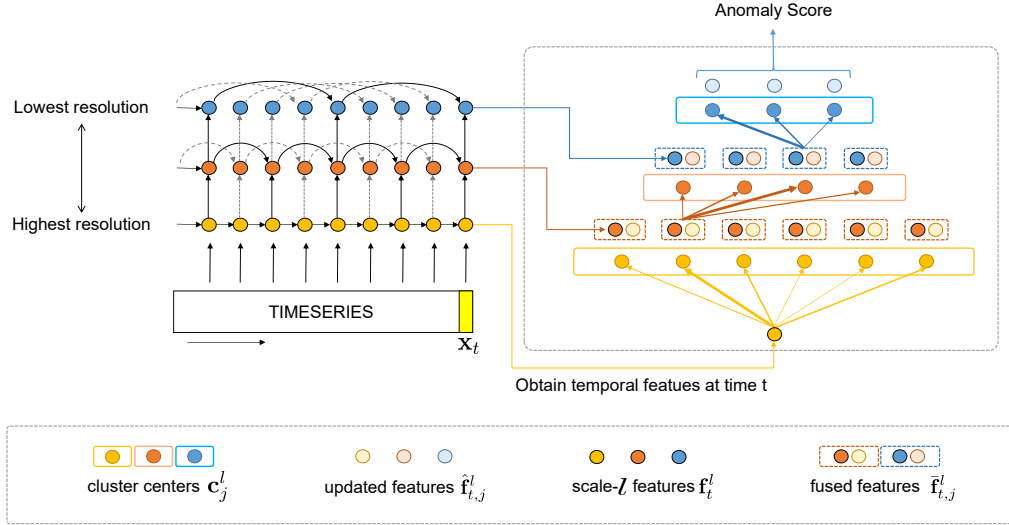

Figure 1: The proposed Temporal Hierarchical One-Class (THOC) network with $L = 3$ layers.

#### 3.1.1   Multiscale Temporal Features

To extract multiscale temporal features from the timeseries, we use an $L$-layer dilated recurrent neural network (RNN) [2] with multi-resolution recurrent skip connections. Other networks capable of extracting multiscale features (such as the WaveNet [20]) can also be used. For simplicity of notations, we drop the subscript $s$ in this section. At time $t$, let the (partial) timeseries that has been observed so far be $\mathbf{x}_{1:t-1}$. For a particular layer $l$, the hidden state $\mathbf{f}_t^l$ of the recurrent cell is:

$$\mathbf{f}_t^l = \begin{cases} \mathcal{F}_{\mathrm{RNN}}(\mathbf{x}_t, \mathbf{f}_{t-s^{(l)}}^l) & \text{if } l = 1, \\ \mathcal{F}_{\mathrm{RNN}}(\mathbf{f}_t^{l-1}, \mathbf{f}_{t-s^{(l)}}^l) & \text{otherwise}, \end{cases} \tag{4}$$

where $\mathcal{F}_{\mathrm{RNN}}$ is any RNN cell (such as the vanilla RNN cell, LSTM or GRU), and $\mathbf{f}_{t-s^{(l)}}^l$ is the input for the skip connection with skip length $s^{(l)}$. The use of skip connections helps model long-term dependencies in the timeseries and alleviates the problem of vanishing gradient. The larger the $s^{(l)}$,

the longer the dependency layer $l$ captures. Similar to the WaveNet [20], an exponentially increasing skip dilation is used here: $s^{(l)} = M_0 \prod_{i=1}^{l-1} M$ (in Figure 1, $M_0 = 1$ and $M = 2$). With multiple layers, the dilated-RNN can extract an abundance of multiscale temporal features, with shorter-term information learned at the lower layers and longer-term information at the upper layers.

### 3.1.2 Fusing the Multiscale Features

Instead of only using features from dilated RNN's last layer, features from intermediate layers may also contain useful information. In this section, we propose a differentiable hierarchical clustering procedure to fuse information from all these different scales.

For a particular scale $l \in \{1, \dots, L\}$, there is a corresponding clustering layer with $K^l$ clusters whose centers are $\{\mathbf{c}_1^l, \dots, \mathbf{c}_{K^l}^l\}$. At time $t$, the inputs to this layer are the outputs from the previous layer $\{\bar{\mathbf{f}}_{t,1}^{l-1}, \dots, \bar{\mathbf{f}}_{t,K^{l-1}}^{l-1}\}$ (which will be detailed in (8)). When $l = 1$, we use the temporal features $\mathbf{f}_t^1$ from (4) as input (and thus $K^0 = 1$). The hierarchical clustering procedure proceeds by alternating the following two steps.

**Step 1 (Assignment)**: Based on the similarities between $\bar{\mathbf{f}}_{t,i}^{l-1}$ and each center $\mathbf{c}_j^l$, we assign $\bar{\mathbf{f}}_{t,i}^{l-1}$ to these centers with probabilities:

$$P_{t,i\to j}^l = P(\bar{\mathbf{f}}_{t,i}^{l-1} \to \mathbf{c}_j^l) = \frac{\exp(\text{score}(\bar{\mathbf{f}}_{t,i}^{l-1}, \mathbf{c}_j^l)/\tau)}{\sum_{k=1}^{K^l} \exp(\text{score}(\bar{\mathbf{f}}_{t,i}^{l-1}, \mathbf{c}_k^l)/\tau)}, \tag{5}$$

where $\tau \in (0, \infty)$ is a temperature parameter, and $\text{score}(\cdot, \cdot)$ is a similarity score function. When $\tau \to \infty$, $\bar{\mathbf{f}}_{t,i}^{l-1}$ is assigned to all the centers with equal probabilities. When $\tau \to 0$, the soft assignment becomes hard. As for the score function, we use the simple cosine similarity:

$$\text{score}(\bar{\mathbf{f}}, \mathbf{c}) = \mathbf{f}^\top \mathbf{c}/(\|\mathbf{f}\| \cdot \|\mathbf{c}\|). \tag{6}$$

Other similarity functions can also be used.

**Step 2 (Update)**: After obtaining the assignment probabilities, all features $\{\bar{\mathbf{f}}_{t,1}^{l-1}, \dots, \bar{\mathbf{f}}_{t,K^{l-1}}^{l-1}\}$ from the lower level are fused and transformed in each cluster $\mathbf{c}_j^l$ as

$$\hat{\mathbf{f}}_{t,j}^l = \sum_{i=1}^{K^{l-1}} P_{t,i\to j}^l \text{ReLU}(\mathbf{W}^l \bar{\mathbf{f}}_{t,i}^{l-1} + \mathbf{b}^l), \quad j = 1, \dots, K^l, \tag{7}$$

where both the weight $\mathbf{W}^l$ and bias $\mathbf{b}^l$ are learned end-to-end with the other model parameters (as will be discussed in Section 3.2). If $l \leq L - 1$, $\hat{\mathbf{f}}_{t,j}^l$ is then concatenated with the corresponding scale-$(l+1)$ feature $\mathbf{f}_t^{l+1}$ from the dilated RNN, and further transformed by a fully-connected layer $\mathcal{F}_{\text{MLP}}$ with output having the same dimension as $\hat{\mathbf{f}}_{t,j}^l$. Note that $\hat{\mathbf{f}}_{t,j}^L$'s are directly used as output at the last layer.

$$\bar{\mathbf{f}}_{t,j}^l = \begin{cases} \mathbf{f}_t^1 & \text{if } l = 0 \\ \mathcal{F}_{\text{MLP}}([\hat{\mathbf{f}}_{t,j}^l; \mathbf{f}_t^{l+1}]) & \text{if } 1 \leq l \leq L - 1 \\ \hat{\mathbf{f}}_{t,j}^L & \text{otherwise (i.e., } l = L) \end{cases}. \tag{8}$$

### 3.2 Multiscale Support Vector Data Description (MVDD)

As in deep SVDD, we measure the difference between features $\{\bar{\mathbf{f}}_{t,j}^L\}$ and centers $\{\mathbf{c}_1^L, \dots, \mathbf{c}_{K^L}^L\}$ at the last layer. As we use the cosine similarity in (6), the cosine distance (i.e., $1 - \text{cosine similarity}$) is used as $d(\bar{\mathbf{f}}_{t,s}^L, \mathbf{c}_k^L)$. Let $\mathcal{W}$ be all the learnable weights in the network. Our objective is:

$$\mathcal{L}_{\text{THOC}} = \frac{1}{NK^L} \sum_{s=1}^N \frac{1}{T_s} \sum_{t=1}^{T_s} \sum_{j=1}^{K^L} R_{t,j,s}^L d(\bar{\mathbf{f}}_{t,j,s}^L, \mathbf{c}_j^L) + \lambda \Omega(\mathcal{W}), \tag{9}$$

where $\Omega(\mathcal{W})$ is the $\ell_2$-regularizer. Note that we have explicitly added back the subscript $s$ for samples. Moreover, while deep SVDD has only one hypersphere, the proposed model involves multiple layers

each with multiple centers. Hence, $R^L_{t,j,s}$ in (9) indicates the degree that current observation $\mathbf{x}_{t,s}$ is associated with center $\mathbf{c}^L_j$.

The following shows that $R^L_{t,j,s}$ can be computed easily in a recursive manner. For simplicity of notations, we drop the subscript $s$ here. At the first layer, the only input is $\mathbf{f}^1_t$, and so $R^1_{t,j} = P^1_{t,1 \to j}$ in (5). For layer $l$, $R^l_{t,j}$ is obtained by a softmax over all centers in the same layer:

$$R^l_{t,j} = \frac{\exp(\tilde{R}^l_{t,j})}{\sum_{i=1}^{K^l} \exp(\tilde{R}^l_{t,i})}, \text{ where } \tilde{R}^l_{t,j} = \begin{cases} P^1_{t,i \to j}, & \text{if } l = 1 \\ \sum_{i=1}^{K^{l-1}} P^l_{t,i \to j} R^{l-1}_{t,i} & \text{if } 1 < l \le L \end{cases}. \tag{10}$$

$\tilde{R}^l_{t,j}$ depends on both the degrees $R^{l-1}_{t,i}$'s that $\mathbf{x}_t$ is associated to centers ($\mathbf{c}^{l-1}_i$'s) in the previous layer $(l-1)$ and the assignment probabilities ($P^l_{t,i \to j}$'s) that the $\bar{\mathbf{f}}^{l-1}_{t,i}$ output from $\mathbf{c}^{l-1}_i$ is assigned to $\mathbf{c}^l_j$.

To allow the centers $\mathbf{c}^l_k$'s in each layer to be as diverse as possible, we add the following loss

$$\mathcal{L}_{\text{orth}} = \frac{1}{L} \sum_{l=1}^{L} \|(\mathbf{C}^l)^\top \mathbf{C}^l - \mathbf{I}\|^2_F, \tag{11}$$

where $\mathbf{C}^l = [\mathbf{c}^l_1 \cdots \mathbf{c}^l_{K^l}]$, $\mathbf{I}$ is the identity matrix, and $\|\cdot\|_F$ is the Frobenius norm. This encourages $\mathbf{c}^l_k$'s to be orthogonal to each other.

Moreover, self-supervision [4], which constructs related auxiliary tasks to aid in the learning of informative features, has recently shown to be an effective unsupervised representation learning method in many real-world applications [4, 11, 7]. For timeseries data, a natural self-supervised learning task is multi-step-ahead prediction. Here, to encourage the learning of useful features at all layers of the dilated-RNN, we use a linear model (with learnable weight $\mathbf{W}^l_{\text{pred}}$ in each layer $l$) to predict $\mathbf{x}_{t,s}$ from the corresponding layer-$l$ hidden state $\mathbf{f}^l_{t-s^{(l)},s}$ at time $t - s^{(l)}$. This leads to the following temporal self-supervision loss (TSS):

$$\mathcal{L}_{\text{TSS}} = \frac{1}{NL} \sum_{s=1}^{N} \sum_{l=1}^{L} \left( \frac{1}{T_s - s^{(l)}} \sum_{t=s^{(l)}+1}^{T_s} \|\mathbf{W}^l_{\text{pred}} \mathbf{f}^l_{t-s^{(l)},s} - \mathbf{x}_{t,s}\|^2 \right). \tag{12}$$

Combining all three losses, we obtain the following *multiscale support vector data description* (MVDD) objective:

$$\mathcal{L}_{\text{total}} = \mathcal{L}_{\text{THOC}} + \lambda_{\text{orth}} \mathcal{L}_{\text{orth}} + \lambda_{\text{TSS}} \mathcal{L}_{\text{TSS}}, \tag{13}$$

where $\lambda_{\text{orth}}$ and $\lambda_{\text{TSS}}$ are tradeoff hyperparameters. All the parameters can be learned in an end-to-end manner. The whole procedure is shown in Algorithm 1.

---

**Algorithm 1** Temporal hierarchical one-class learning (THOC).

---

**Input:** timeseries $\mathbf{X}_s = (\mathbf{x}_{1,s}, \mathbf{x}_{2,s}, \ldots, \mathbf{x}_{T_s,s})$; number of centers $\{K^l\}$; skip lengths $\{s^{(l)}\}$.
1: **repeat**
2:      feed $\mathbf{x}_{t,s}$ into the $L$-layer dilated-RNN and obtain $\{\mathbf{f}^l_t\}$ from each layer;
3:      **for** layer $l = 1, \ldots, L$ **do**
4:          obtain the $l$th clustering layer's input $\{\bar{\mathbf{f}}^{l-1}_{t,i}\}_{i=1,\ldots,K^{l-1}}$ by (8) where $K^0 = 1$;
5:          compute probabilities $\{P^l_{t,i \to j}\}_{i=1,\ldots,K^{l-1}, j=1,\ldots,K^l}$ from (5);
6:          compute $\{R^l_{t,j}\}_{j=1,\ldots,K^l}$ for each cluster center at layer $l$ from (10);
7:          update and obtain output features $\{\hat{\mathbf{f}}^l_{t,j}\}_{j=1,\ldots,K^l}$ from (7);
8:      **end for**
9:      minimize MVDD objective in (13) by the Adam optimizer;
10: **until** convergence.

---

With the trained model, let the observation for an unseen timeseries $\mathbf{X}$ at time $t$ be $\mathbf{x}_t$. As in (9), we define its anomaly score as: $\text{AnomalyScore}(\mathbf{x}_t) = \sum_{j=1}^{K^L} R^L_{t,j} \cdot d(\bar{\mathbf{f}}^L_t, \mathbf{c}^L_j)$. Given a predefined threshold $\delta$, we then label $\mathbf{x}_t$ as abnormal if $\text{Anomaly Score}(\mathbf{x}_t) > \delta$, and normal otherwise.

# 4 Experiments

In this section, we demonstrate the performance of the proposed model on a number of commonly used benchmark timeseries data sets.

## 4.1 Data Sets

The following timeseries data sets are used:

(i) *2D-gesture* [9], which records the X-Y coordinate sequences of hand gestures in a video;

(ii) *Power demand*[9], which contains a year of power demand at a Dutch research facility;

(iii) *KDD-Cup99* data from the DARPA'98 Intrusion Detection Evaluation Program [14]. It contains around seven million network traffic connection records over a 7-week period. A connection is a sequence of TCP packets. Each record is labeled as either normal or attack;

(iv) Secure Water Treatment (*SWaT*) data [18], which is collected from a water treatment testbed over 11 days. 36 attacks were launched during the last 4 days of the collection process. These attacks were launched with different intents and diverse lasting durations (from a few minutes to an hour);[1]

(v) Mars Science Laboratory rover (*MSL*); and

(vi) Soil Moisture Active Passive satellite (*SMAP*) data: Both *MSL* and *SMAP* are public data sets from NASA [8]. They contain telemetry anomaly data derived from the Incident Surprise Anomaly (ISA) reports of spacecraft monitoring systems. Each data set has a training and a testing set. Anomalies in the testing set are labeled, while the training set contains unlabeled anomalies.

As will be seen in section 4.2, some of the baselines are designed for non-temporal data. Thus, a sliding window is needed to convert the timeseries to fixed-length input. Specifically, the raw timeseries data is partitioned into fixed-length sequences (80 for *2D-gesture* and *power-demand*, and 100 for the others) by using a sliding window (with stride 100 for *MSL* and *SMAP*, and 1 for the others). To allow fair comparison, we employ the same data preprocessing for all methods.

Table 1: Statistics of the data sets used.

|  | dim | length | #training | #validation | #testing |
|---|---|---|---|---|---|
| *2D-gesture* | 2 | 80 | 8,170 | 420 | 2,420 |
| *power-demand* | 1 | 80 | 18,145 | 4,786 | 10,000 |
| *KDD-Cup99* | 34 | 100 | 56,139 | 24,601 | 24,602 |
| *SWaT* | 51 | 100 | 47,420 | 22,396 | 22,396 |
| *MSL* | 55 | 100 | 40,721 | 17,396 | 73,629 |
| *SMAP* | 25 | 100 | 94,528 | 40,455 | 427,517 |

For *2D-gesture*, *power-demand*, *KDD-Cup99*, and *SWaT*, the raw data set has only a training set and a test set. To allow model selection and hyperparameter tuning, we use part of the provided test set for validation. For *MSL* and *SMAP*, we follow the setting in [28], and hold out 30% of the training data as validation set. A summary of the resultant data set statistics is shown in Table 1.

## 4.2 Baselines for Comparison

The proposed model is compared with the following groups of anomaly detection algorithms.[2] The first group contains anomaly detectors for general multivariate data. These include traditional one-class classifiers: (i) local outlier factor (LOF) [1], (ii) one-class SVM (OC-SVM) with RBF kernel [29], and (iii) isolation forest [16]), and the recent deep learning models of (i) deep SVDD [22], (ii) AnoGAN [24], and (iii) deep autoencoding Gaussian mixture model (DAGMM) [34]. The second group contains deep-network-based anomaly detectors for timeseries data. These include (i) the

Table 2: Precision (prec), recall (rec) and F1 score results (as %) on various data sets. The number in brackets after the F1 value is the rank of the method. The smaller the better.

| | 2D-gesture | | | power-demand | | | KDD-Cup99 | | | SWaT | | | avg rank |
|---|---|---|---|---|---|---|---|---|---|---|---|---|---|
| | prec | rec | F1 | prec | rec | F1 | prec | rec | F1 | prec | rec | F1 | |
| LOF | 27.82 | 87.21 | 42.18 (8) | 15.29 | 28.13 | 19.81 (9) | 95.38 | 99.55 | 97.42 (11) | 76.97 | 98.36 | 86.36 (7) | 8.75 |
| OC-SVM | 65.50 | 25.57 | 36.78 (14) | 12.40 | 60.43 | 20.58 (8) | 95.25 | 99.92 | 97.53 (10) | 99.47 | 61.47 | 75.98 (13) | 11.25 |
| iso forest | 28.54 | 68.04 | 40.22 (10) | 7.85 | 89.77 | 14.44 (13) | 96.85 | 99.38 | 98.10 (7) | 99.00 | 74.47 | 85.00 (9) | 9.75 |
| deep SVDD | 26.26 | 64.53 | 37.32 (13) | 11.51 | 64.74 | 19.54 (10) | 89.83 | 100.0 | 94.64 (14) | 97.68 | 71.88 | 82.82 (11) | 12 |
| AnoGAN | 57.85 | 46.50 | 51.55 (4) | 20.28 | 44.41 | 28.85 (5) | 93.11 | 99.93 | 96.40 (12) | 99.01 | 77.01 | 86.64 (5) | 6.5 |
| DAGMM | 25.66 | 80.47 | 38.91 (12) | 34.37 | 41.72 | 37.69 (4) | 96.12 | 99.70 | 97.86 (8) | 90.60 | 80.72 | 85.38 (8) | 8.0 |
| EncDec-AD | 24.88 | 100.0 | 39.85 (11) | 13.98 | 54.20 | 22.22 (6) | 89.74 | 99.50 | 94.37 (13) | 93.69 | 63.31 | 75.56 (14) | 11 |
| LSTM-VAE | 36.62 | 67.76 | 47.54 (6) | 8.00 | 56.66 | 14.03 (14) | 98.84 | 98.09 | 98.47 (3) | 98.39 | 77.01 | 86.39 (6) | 7.25 |
| MadGAN | 29.41 | 76.40 | 42.47 (7) | 13.20 | 60.57 | 21.67 (7) | 96.73 | 99.55 | 98.12 (6) | 98.72 | 77.60 | 86.89 (2) | 5.5 |
| BeatGAN | 55.11 | 45.33 | 49.74 (5) | 8.04 | 76.58 | 14.56 (12) | 97.54 | 98.94 | 98.23 (5) | 88.37 | 76.41 | 81.95 (12) | 8.5 |
| OmniAnomaly | 27.70 | 79.67 | 41.11 (9) | 8.55 | 78.73 | 15.42 (11) | 97.63 | 99.69 | 98.65 (2) | 99.01 | 77.06 | 86.67 (4) | 6.5 |
| MSCRED | 61.26 | 59.11 | 60.17 (2.5) | 55.80 | 34.32 | 42.50 (3) | 97.31 | 99.43 | 98.36 (4) | 98.43 | 77.69 | 86.84 (3) | 3.125 |
| CVDD | 56.05 | 64.95 | 60.17 (2.5) | 49.65 | 38.36 | 43.30 (2) | 96.37 | 98.75 | 97.54 (9) | 97.33 | 73.21 | 83.56 (10) | 5.875 |
| THOC | 54.78 | 75.00 | **63.31** (1) | 61.50 | 36.34 | **45.68** (1) | 98.20 | 99.54 | **98.86** (1) | 98.08 | 79.94 | **88.09** (1) | 1.0 |

encoder-decoder scheme for anomaly detection (EncDec-AD) [17], (ii) LSTM-VAE [21], (iii) MAD-GAN [13], (iv) BeatGAN [33], (v) OmniAnomaly [28], and (vi) multi-scale convolutional recurrent encoder-decoder (MSCRED) [31]. As additional baselines, we also compare with the context vector data description (CVDD) [23], which is a recent deep network for text-specific anomaly detection using distributed word representations. We adapt the CVDD for timeseries by using the dilated RNN for feature representation. Hyperparameters of the proposed method and baselines are selected based on the F1 value on validation set. Detailed experimental settings can be found in Appendix B.

For performance evaluation, we use the standard metrics of (i) precision; (ii) recall; and (iii) F1 score. On the MSL and SMAP data sets, we follow [30, 28] and adjust the anomaly detection results as follows: If a point in a contiguous anomalous segment is detected correctly, all anomalies in the same segment are also considered to have been correctly detected. This adjustment is justified by the observation that the time point causing the anomaly does not need to be exactly detected in practice.

### 4.3 Results on *2D-gesture*, *power-demand*, *KDD-Cup99*, and *SWaT*

Table 2 shows the results on *2D-gesture*, *power-demand*, *KDD-Cup99*, and *SWaT*. As can be seen, all methods perform better on *KDD-Cup99* and *SWaT*, which have larger training sets and relatively simpler temporal dynamics. Moreover, not surprisingly, anomaly detectors for general multivariate data do not perform well on timeseries, as they do not model the underlying temporal dependency well even with the use of a sliding window. Timeseries anomaly detectors based on deep generative models (EncDec-AD, LSTM-VAE, AnoGAN, MAD-GAN, BeatGAN and OmniAnomaly) do not fare much better in general. As discussed in [28], EncDec-AD may have difficulty in encoding all the information in the timeseries, while the other generative models often lack sufficient consideration of the temporal dependence among stochastic variables. Moreover, GAN-based models are usually harder to train since they can easily suffer from the mode collapse and convergence problems [10].

The recent MSCRED and CVDD achieve good overall performance. However, one limitation of MSCRED is that it relies on the covariance among different dimensions in the multivariate timeseries. This measures only the linear dependency among dimensions, and may be problematic when the underlying interactions are complex and nonlinear. As for CVDD, all its hyperspheres are organized in a single layer, and so cannot capture multi-scale temporal characteristics from the data.

The proposed THOC model outperforms all the baselines on all data sets. The hierarchical structure with multiple hyperspheres in each layer efficiently fuses the multiscale temporal information, allowing the capture of complex temporal dynamics in the timeseries data.

## 4.4 F1 Results on *MSL* and *SMAP*

Following [28], we only report the F1-values on these two data sets. Figure 2 shows the results (results of DAGMM, EncDec-AD, LSTM-VAE, and OmniAnomaly are provided by the authors of [28]). Again, the proposed THOC model achieves the best performance.

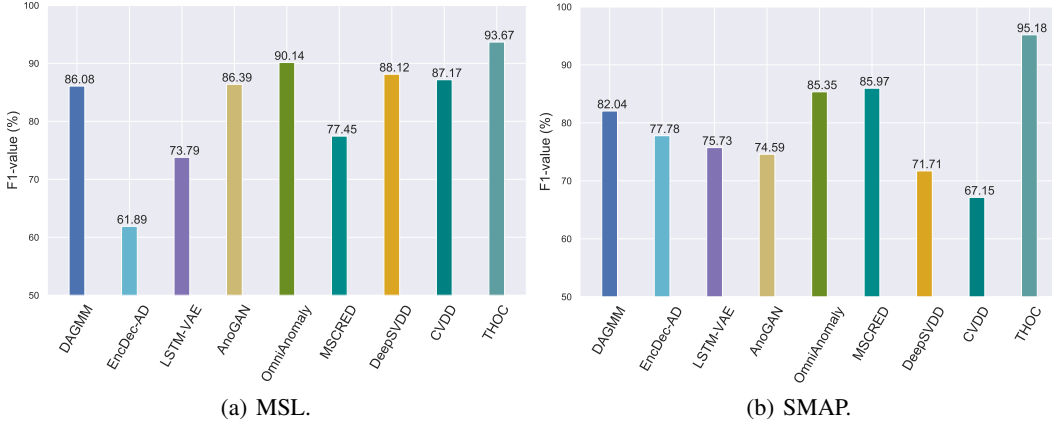

(a) MSL.                                    (b) SMAP.

Figure 2: F1 results on the MSL and SMAP data sets.

## 4.5 Ablation Study

In this section, we perform ablation studies on the following aspects of the proposed THOC: (i) How to represent the timeseries? (ii) Use a hierarchical structure as in THOC or a flat structure as in deep SVDD? (iii) Effectiveness of the two loss components $\mathcal{L}_{\text{orth}}$ and $\mathcal{L}_{\text{TSS}}$ in (13).

### 4.5.1 Timeseries Representation and Hierarchical Structure

Recall that we use a hierarchical timeseries representation, and multiple hyperspheres are used to capture the temporal dynamics at each resolution. In this experiment, we demonstrate their effectiveness on the *2D-gesture* data set.

First, instead of exploiting the multiscale structure in timeseries, we consider the alternative of using the deep SVDD, a flat one-class classifier. We use as input the features (denoted "RNN-top") from the top layer of dilated RNN. To allow a fair comparison, the self-supervision loss ($\mathcal{L}_{\text{TSS}}$ in (12)) is also used in the joint end-to-end training of the RNN representation and deep SVDD. As deep SVDD uses only one hypersphere to capture the samples' normal behavior, $\mathcal{L}_{\text{orth}}$ in (11) is no longer needed. We also experiment with a recent general-purpose unsupervised timeseries representation proposed in [5]. Following its paper title, this is denoted USRL. The USRL representation is obtained from an encoder with causal dilated convolutions (in this experiment, we use a three-layer casual convolution network). This is trained with a triplet loss that encourages the representation of the whole timeseries to be close to those of its subseries. Note that while the proposed model involves a $\mathcal{L}_{\text{TSS}}$ component in the training objective, USRL does not utilize self-supervised learning.

As the proposed THOC network uses multiple hyperspheres for anomaly detection, we also feed the single-resolution timeseries representations (RNN-top and USRL) to the proposed MVDD with multiple hyperspheres. As in THOC, $\mathcal{L}_{\text{orth}}$ is used to encourage diversity of the hypersphere centers.

To further investigate the effectiveness of having multiple hyperspheres in each layer in THOC, we additionally experiment with a THOC variant which uses only one hypersphere in each layer (i.e., $K^l = 1$ for $l = 1, 2, \ldots, L$). As each layer has only one hypersphere, the $\mathcal{L}_{\text{orth}}$ component in the objective becomes unnecessary.

Results are shown in Table 3. As can be seen, the proposed THOC, which uses all levels of the dilated RNN's multiscale features, has much better performance than the single-resolution timeseries representations of RNN-top and USRL. This demonstrates that flat models (even with multiple hyperspheres) have limited capabilities in modeling the timeseries.

Table 3: Comparison of variants using different flat and hierarchical timeseries representations.

| | | | $\mathcal{L}_{\mathrm{orth}}$ | $\mathcal{L}_{\mathrm{TSS}}$ | prec | recall | F1 |
|---|---|---|---|---|---|---|---|
| flat | one hypersphere | RNN-top | × | ✓ | 31.67 | 75.70 | 44.66 |
| | | USRL | × | × | 59.49 | 27.10 | 37.24 |
| | multiple hyperspheres | RNN-top | ✓ | ✓ | 41.32 | 68.93 | 51.66 |
| | | USRL | ✓ | × | 50.80 | 45.40 | 47.95 |
| hierarchical | one hypersphere | THOC-variant | × | ✓ | 53.27 | 60.98 | 56.86 |
| | multiple hyperspheres | THOC | ✓ | ✓ | 54.78 | 75.00 | **63.31** |

Though USRL demonstrates encouraging performance on timeseries classification [5], it does not perform well on timeseries anomaly detection. Recall that in training the USRL, the triplet loss encourages the representation of the whole timeseries to be close to those of its subseries. This implicitly assumes that the whole timeseries contains no anomalous subseries, and is thus not suitable for anomaly detection. Besides, though the dilated convolution network used in USRL also extracts multi-scale features (similar to the dilated RNN in the proposed THOC), the final USRL representation is extracted only from its top layer. This suffers from a loss of fine-grained temporal information as the RNN-top representation.

Comparing the use of one versus multiple hyperspheres, it can be seen that using multiple hyperspheres is more advantageous in both the flat and hierarchical models. This verifies that the complex temporal dynamics of real-world timeseries cannot be sufficiently captured by one single hypersphere. By combining temporal representations from multiple resolutions and using hierarchical fusion with multiple hyperspheres, THOC achieves the best F1-value of 63.31%.

### 4.5.2 Effectiveness of $\mathcal{L}_{\mathrm{orth}}$ and $\mathcal{L}_{\mathrm{TSS}}$

In this experiment, we consider the effectiveness of $\mathcal{L}_{\mathrm{orth}}$ and $\mathcal{L}_{\mathrm{TSS}}$ by dropping one or both components from the objective in (13). Results are shown in Table 4. As can be seen, both $\mathcal{L}_{\mathrm{orth}}$ and the $\mathcal{L}_{\mathrm{TSS}}$ are indeed important. Without the orthogonal loss, the centers may be very similar or even duplicate; without the self-supervised loss (which is based on timeseries prediction), the model may fail to capture temporal dependencies, which are essential for a proper representation of timeseries data.

Table 4: Effectiveness of $\mathcal{L}_{\mathrm{orth}}$ and $\mathcal{L}_{\mathrm{TSS}}$.

| $\mathcal{L}_{\mathrm{orth}}$ | $\mathcal{L}_{\mathrm{TSS}}$ | prec | recall | F1 |
|---|---|---|---|---|
| × | × | 52.22 | 24.77 | 33.60 |
| ✓ | × | 34.00 | 67.29 | 45.17 |
| × | ✓ | 42.08 | 57.71 | 48.67 |
| ✓ | ✓ | 54.78 | 75.00 | **63.31** |

## 5 Conclusion

In this paper, we introduced an improved deep model for timeseries anomaly detection. The proposed Temporal Hierarchical One-Class (THOC) network is based on a set of hierarchical structured hyperspheres. The solution uses a probabilistic relevance on cluster centers to help the model access the whole temporal history. A center orthogonality loss and a temporal self-supervision loss are also introduced for improved feature representation. We experimentally demonstrate the effectiveness of each component in our model. Comparisons with state-of-the-art baselines on a number of real-world timeseries benchmarks demonstrate that the proposed model consistently outperforms existing timeseries anomaly detection methods.

## Broader Impact

Timeseries anomaly detection is important for complex cyber-physical systems such as power plants, data centers, and smart factories. By monitoring the system's real-time working conditions, timeseries anomaly detection techniques can automatically detect the abnormal status of the system such that potential risks and financial loss can be avoided. This is very beneficial to the development of social economy and urban security.

## Footnotes

[1]The *KDD-Cup99* and *SWaT* data are downsized by a downsampling rate of 10:1.

[2]Source codes of the baselines are downloaded from the web.

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
