[Supplementary Material]

# Supplementary Material: Timeseries Anomaly Detection using Temporal Hierarchical One-Class Network

**Lifeng Shen[1], Zhuocong Li[2], James T. Kwok[1]**
[1] Department of Computer Science and Engineering,
Hong Kong University of Science and Technology, Hong Kong
{lshenae,jamesk}@cse.ust.hk
[2] Cloud and Smart Industries Group, Tencent, China
zhuocongli@tencent.com

## A  Data sets

*2D-gesture* and *Power demand* [3] can be downloaded from the link `https://www.cs.ucr.edu/~eamonn/discords/`.

*KDD-Cup99* dataset can be obtained from `http://kdd.ics.uci.edu/databases/kddcup99/kddcup99.html`.

*SWaT* is from `https://itrust.sutd.edu.sg/testbeds/secure-water-treatment-swat/`.

*MSL* (Mars Science Laboratory rover) [2] and *SMAP* [A] [2] (Soil Moisture Active Passive satellite) are downloaded from `https://s3-us-west-2.amazonaws.com/telemanom/data.zip` .

## B  Experimental Setting

For LOF [1], the number of neighbors is selected from {1, 3, 5, 12}. For one-class SVM [8], the RBF kernel is used. Its inverse length $\gamma$ is selected from the {0.0001, 0.001, 0.01, 0.1, 0.5}. $\nu$ is another hyperparameter in the OC-SVM, which is selected from {0.1, 0.2, 0.6}. For the isolation forest [6]), the number of tree is selected from {25, 100}. For DAGMM [10], we use its default hyperparameters. For GAN-based baselines (AnoGAN [7], MAD-GAN [5], BeatGAN [9]), we use a sliding window to extract recent history information for prediction. The window length is 80 on *2D-gesture* and *power-demand*, and 100 on the other data sets. For samples tested at multiple windows, we use its average anomaly score over the windows as the final evaluation score.

LOF, OC-SVM and isolation forest are implemented with the Scikit-learn library. The other baselines are downloaded from the following:

- DAGMM: `https://github.com/danieltan07/dagmm`
- EncDec-AD: `https://github.com/KDD-OpenSource/DeepADoTS`
- LSTM-VAE: `https://github.com/SchindlerLiang/VAE-for-Anomaly-Detection`
- AnoGAN: `https://github.com/LeeDoYup/AnoGAN-tf`
- BeatGAN: `https://github.com/Vniex/BeatGAN`
- MadGAN: `https://github.com/LiDan456/MAD-GANs`
- OmniAnomaly: `https://github.com/NetManAIOps/OmniAnomaly`
- MSCRED: `https://github.com/wxdang/MSCRED`

- CVDD: `https://github.com/lukasruff/CVDD-PyTorch`
- Deep SVDD: `https://github.com/lukasruff/Deep-SVDD`

In the proposed model, we use a three-layer dilated RNN with $\ell_2$-regularization (with regularization parameter $10^{-6}$). The number of hidden units is chosen from $\{32, 64, 84\}$. For the number of centers in each layer $\{K^l\}$, empirically we found that simply using a constant or decreasing sequence ($K^1 \geq \cdots \geq K^L$) achieve good performance. Specifically, we select $\{K^l\}$ from $\{\{6,6,6\}, \{12, 6, 1\}, \{12, 6, 4\}, \{18, 6, 1\}, \{18, 12, 4\}, \{18, 12, 6\}, \{32, 12, 6\}\}$, and $\{s^{(1)}, \ldots, s^{(L)}\}$ from $\{\{1,2,4\}, \{1,4,8\}, \{1,4,12\}, \{1,4,16\}\}$. Centers in each layer are initialized by $k$-means clustering on the hidden states. $\lambda_{\mathrm{orth}}$ and $\lambda_{\mathrm{TSS}}$ are selected from $\{0.01, 0.1, 1, 10, 100\}$. We use the Adam optimizer [4]. The initial learning rate is 0.01 for *2D-gesture*, *Power demand* and *KDD-Cup99*; and 0.001 for the other datasets. This is decayed by a factor of 0.65 after every 20 epochs. The initial value of the $1/\tau$ in Eq.(5) is selected from $\{0.01, 1, 10, 20\}$, and is increased by a factor of 1.5 every 5 epochs, until a maximum of 300 is reached. The batch-size is selected from $\{32, 64, 128\}$. The experiments are run on the PyTorch platform using a GeForce GTX1080-Ti11G GPU.