[Reviews · NeurIPS 2020]

Review 1

Summary and Contributions: This paper presents an unsupervised attention based feed-forward neural network which uses multi-resolution features extracted by dilated RNNs as inputs. SVDD loss is used as an anomaly objective function. In order to encourage neural networks to learn informative features, an orthogonal loss as well as self supervised (an auto-regression loss) loss are also used. The empirical studies on 6 popular time series datasets justified the effectiveness of the proposed technique. The major contribution is enable the model to consider mutli-resolution features in time series anomaly detection problem.

Strengths: The major strength of this paper is their models' performance on 6 datasets. Their model outperforms all 13 baselines by a large margin.

Weaknesses: One potential weakness is lacking of novelty. The neural network framework can be separated into 3 parts: multi-resolution feature extraction module using dilated RNNs; Features fusion module using attention mechanism; Anomaly loss using SVDD. All three modules are known techniques with little technical improvements. The presentation of this paper needs to be improved. For example, section 3.1.2 should cite attention mechanism rather than re-invent some technical terms. Also authors do not mention how multi-resolution features extracted from dilated RNNs have been inputed into fusion module.

Correctness: Both claims and method are plausible. Empirical experiments show outstanding performance.

Clarity: Generally the paper is well written and easy to follow. However as stated in previous section, several key parts are not very clear. Also, authors should consider to summarize contributions in introduction.

Relation to Prior Work: Yes.

Reproducibility: Yes

Additional Feedback: (1) The justification of model design is not fully convincing. For instance, this work employs dilated RNN to extract multiscale temporal features. However, similar works (e.g., wavenet, MSCRED) have been developed to address this issue. It is not clear why dilated RNN is a valid choice. (2) Instead of using attention mechanism together with orthogonal loss as well as auto-regression loss, authors may consider to use self-attention mechanism to simplify their framework. ============================== My concerns have been partially addressed. Therefore, I am happy to raise my score.


Review 2

Summary and Contributions: The paper addresses the task of unsupervised anomaly detection in time-series using a multi-scale RNN to extract temporal features and a hierarchical network to output anomaly scores. The authors build on Deep SVDD by extending the single hypersphere case to cover multi modalities and multi-level features. They also include a self-supervised loss (reconstruction loss). The proposed model outperforms all other competitors on 4 anomaly-detection datasets.

Strengths: The performance and the ablation study section demonstrate the importance of using both a hierarchical structure and additional losses.

Weaknesses: In the "Broader Impact" section the authors state: "This work extends the one-class classification method to time series". As the authors themselves previously mentioned: "One-class classification [...] The idea is that by assuming that most of the training data are normal, their characteristics are captured and learned by a model. An outlier is then detected when the current observation from the time series cannot be well-fitted by the model". This is a very generic description which can potentially fit any unsupervised anomaly detection method (e.g. an autoencoder trained for reconstruction on normal data only would fit it), rendering the first statement not true. I think the authors should remove it.

Correctness: The claims and empirical methodology seem correct.

Clarity: The use of multiple indices in section 3 makes it really hard to follow the method workflow. The use of "s" for both the sample and the skip dilation doesn't help in that sense. Figure 1 (right) is not clear, as it seems to indicate a tree structure in the hierarchical network. I encourage the authors to replace it to enhance the overall clarity.

Relation to Prior Work: The authors clearly state how their work differs from Deep SVDD, which can be considered here as a baseline.

Reproducibility: Yes

Additional Feedback: After the rebuttal I am still positive toward the paper so my rating is unchanged.


Review 3

Summary and Contributions: This paper proposes a temporal hierarchical one-class neural network for time series anomaly detection based on the proposed multiscale support vector data description (MVDD) based objective function. Extending the support vector data description (SVDD) for the anomaly detection from dynamic data is an important and interesting problem. In order to show the advantages of the proposed approach, this paper conducts thorough experiments on various datasets and compares the proposed approach with many basic and advanced baselines.

Strengths: This paper targets on an important problem, time series anomaly detection, and extends the objective function for static data from deep SVDD to the time series data by proposing MVDD. The proposed approach beats a number of baselines for time series anomaly detection in terms of precision, recall, and F1.

Weaknesses: I am confused about the MVDD part of the proposed approach. For deep SVDD, only one center \mathbf{c} is used to represent the normal class, which is easy to understand. However, for the proposed approach, in each layer of RNN, there are K^l clusters. I cannot fully understand the motivation of this setting. It also means there are so many hyper-parameters (K^l) in the proposed neural network. I think at least, the parameter sensitivity regarding K^l should be evaluated in experiments. Meanwhile, we can also observe from the ablation study shown in Table 5 that when only using the MVDD-based loss, the performance is poor, worse than most of the baselines. It seems that the self-supervised and orthogonal losses play an important role for the anomaly detection.

Correctness: Since the proposed MVDD is different from the commonly used form of SVDD, it would be better to provide the theoretical analysis about MVDD on mapping anomalies outside the hypersphere.

Clarity: Overall, the paper is well written. The notations are a little bit complicated and make the equations not easy to follow. It would be better to put the Tables 2 and 4 in the same place for clarity.

Relation to Prior Work: Yes

Reproducibility: No

Additional Feedback: As pointed out by R1, this paper conducts a good empirical study, so I am willing to increase the overall score.

[Author Response · NeurIPS 2020]

We thank the reviewers for their insightful feedback. In the following, we address their concerns and questions.

**[R1]** "**Lacking of novelty... All three modules are known techniques with little technical improvements**": As will
be discussed in more detail in the sequel, there might be some significant misunderstanding by the reviewer on modules
2 and 3 (i.e., module 2 is not based on attention; module 3 is not standard deep SVDD loss). Moreover, module 1 is
not our main novelty, though for timeseries it is essential to use some model to extract multiscale features. Our main
novelty is on developing a hierarchical one-class model for timeseries, which allows diverse patterns of multiscale
features from normal timeseries to be captured.

"**Features fusion module using attention mechanism ... section 3.1.2 should cite attention mechanism rather
than re-invent some technical terms.**" : There is a big misunderstanding. The proposed method is more similar to
clustering than attention, both of which are based on the notion of similarity. If equation (5) is viewed as attention
as suggested, we would have $\bar{\mathbf{f}}_t^{l-1}$ as the query, and $\{\mathbf{c}_j^l\}_{j=1}^{K^l}$ as the keys (as normalization is w.r.t. the $K^l$ $\mathbf{c}_j^l$'s), and
equation (7) would correspond to the attention output. However, note that the summation in (7) involves $K^{l-1}$ (instead
of $K^l$) terms, and so obviously this cannot be attention.

Indeed, as mentioned at line 116, this should be understood
as a *hierarchical clustering* procedure. The $\bar{\mathbf{f}}_t^{l-1}$'s are clus-
tered by their similarities w.r.t. (learnable) centers $\mathbf{c}_j^l$'s using
(5), to form the higher-level $\hat{\mathbf{f}}_{t,i}^l$'s in (7). At the intermediate

| | MSL (%) | | | SMAP (%) | | |
|---|---|---|---|---|---|---|
| | prec | rec | F1 | prec | rec | F1 |
| attention | 90.66 | 87.21 | 88.90 | 79.70 | 66.35 | 72.41 |
| self-attention | 82.49 | 93.33 | 87.77 | 94.55 | 56.05 | 70.38 |
| proposed | 92.85 | 94.51 | **93.67** | 91.33 | 99.36 | **95.18** |

layers, $\hat{\mathbf{f}}_{t,j}^l$ is further merged with the multiscale feature $\mathbf{f}_t^{l+1}$ (extracted by dilated RNN) to form $\bar{\mathbf{f}}_{t,j}^l$.

For experiments, we add two baselines that use attention to obtain $\hat{\mathbf{f}}_{t,i}^l$ (and uses a one-layer MLP to fuse the query
and learned attention vector):[1] (i) attention, using query $\bar{\mathbf{f}}_t^{l-1}$, keys $\{\mathbf{c}_j^l\}_j$, and values $\{\mathbf{c}_j^l\}_j$; (ii) self-attention, using
query $\bar{\mathbf{f}}_t^{l-1}$, keys $\{\bar{\mathbf{f}}_\tau^{l-1}\}_{\tau<t}$, and values $\{\bar{\mathbf{f}}_\tau^{l-1}\}_{\tau<t}$. Because of lack of time, experiments are only run on the MSL and
SMAP data sets. The table above shows that the proposed model performs best. The discussion above and experimental
results will be added to the final version.

"**Anomaly loss using SVDD**": Deep SVDD uses only one center and one layer, while we have multiple centers ($\mathbf{c}_j^L$'s)
and multiple layers. Besides the simple extension that sums over all $\mathbf{c}_j^L$'s, the key challenge is on what contribution
of each $\bar{\mathbf{f}}_{t,j,s}^L$ at the last layer should be compared with each $\mathbf{c}_j^L$. We propose to aggregate the contributions from the
features $\bar{\mathbf{f}}_{t,j,s}^l$'s at all layers via an efficient recursive computation of $\{\tilde{R}_{t,j}^l\}$.

"**Authors do not mention how features extracted from dilated RNNs have been inputed into fusion module**": The
reviewer might have overlooked parts of the paper. Indeed, these have been discussed at lines 115-116 and 126-128.

"**Not clear why dilated RNN is a valid choice**": As suggested by the reviewer, the dilated RNN can be replaced by
any other model that can extract multiscale features. We used the dilated RNN only as an example model. As discussed
above, this part is not our main novelty. Our key novelty is on how to model the diverse patterns of multiscale features
extracted from timeseries (by the clustering and anomaly detection modules, and the combination of losses).

**[R3]** "**Broader Impact... use of multiple indices... Figure 1 (right) is not clear**": Thanks for your suggestion. We
will improve our description on one-class learners, simplify notations and make the graph clearer in the final version.

**[R4]** "**In each layer, there are $K^l$ clusters... motivation**": In SVDD, the ball is used to enclose most of the normal
patterns. However, as mentioned in line 66-67, real-world timeseries data may have complex characteristics and so
multiple balls can better model the diverse normal temporal behaviors at each resolution (this is similar to using the
more flexible Gaussian mixture over a single Gaussian in other machine learning models).

"**So many hyper-parameters ($K^l$) ... sensitivity regarding $K^l$**": Important hyper-parameters in our network mainly
are $\lambda_{\mathrm{orth}}$ and $\lambda_{\mathrm{TSS}}$. For $K^l$, to construct a hierarchical clustering structure, we empirically found that simply using a
constant or a decreasing setting such as (6, 6, 6) or (18, 12, 6) for $(K^1, \ldots, K^L)$ can achieve a good performance. Thus,
we use a relative small search space to find a good (suboptimal) hyperparameter setting for $K^l$.

"**Self-supervised and orthogonal losses**": These two losses are important. Without the orthogonal loss, the centers
may be very similar or even duplicate; without the self-supervised loss (which is based on timeseries prediction), the
model may fail to capture temporal dependencies, which are essential for a proper representation of timeseries data.

"**Theoretical analysis about MVDD**": As in deepSVDD, when we use the soft-boundary SVDD objective in each
hypersphere, the proof of their Proposition 4 can be easily extended to our model, and the $\nu$-property holds. Here, we
give a brief proof sketch: Define $d_{i,j} = \|\mathrm{NN}_j(x_i; \mathcal{W}_j) - \mathbf{c}_j\|^2$ for $i = 1, 2 \ldots, N$ and $j = 1, 2 \ldots, K^L$. The number
of outliers for the $j$th hypersphere is $N_{\mathrm{out}}^j = |\{i : d_{i,j} > r^2\}|$. It can be shown that the soft-boundary objective of our
model can be rewritten as $\left(1 - \frac{\sum_{j=1}^{K^L} R_j^L N_{\mathrm{out}}^j}{\nu N}\right) r^2$. The remaining steps are similar to that in the proof of deepSVDD.

## Footnotes

[1]As in the literature on attention models, we will specify the query, keys, and values for each baseline.


[Meta-Review · NeurIPS 2020]

This paper received 3 reviews. The reviewers appreciated the extension of support vector data description (SVDD) to temporal data, with interesting multi-scale feature analysis for anomaly detection in time series. The empirical results are strong. After reading the authors' rebuttal, the reviewers converged on a unanimous accept rating for the paper.